# ConsistentAvatar: Learning to Diffuse Fully Consistent Talking Head Avatar with Temporal Guidance

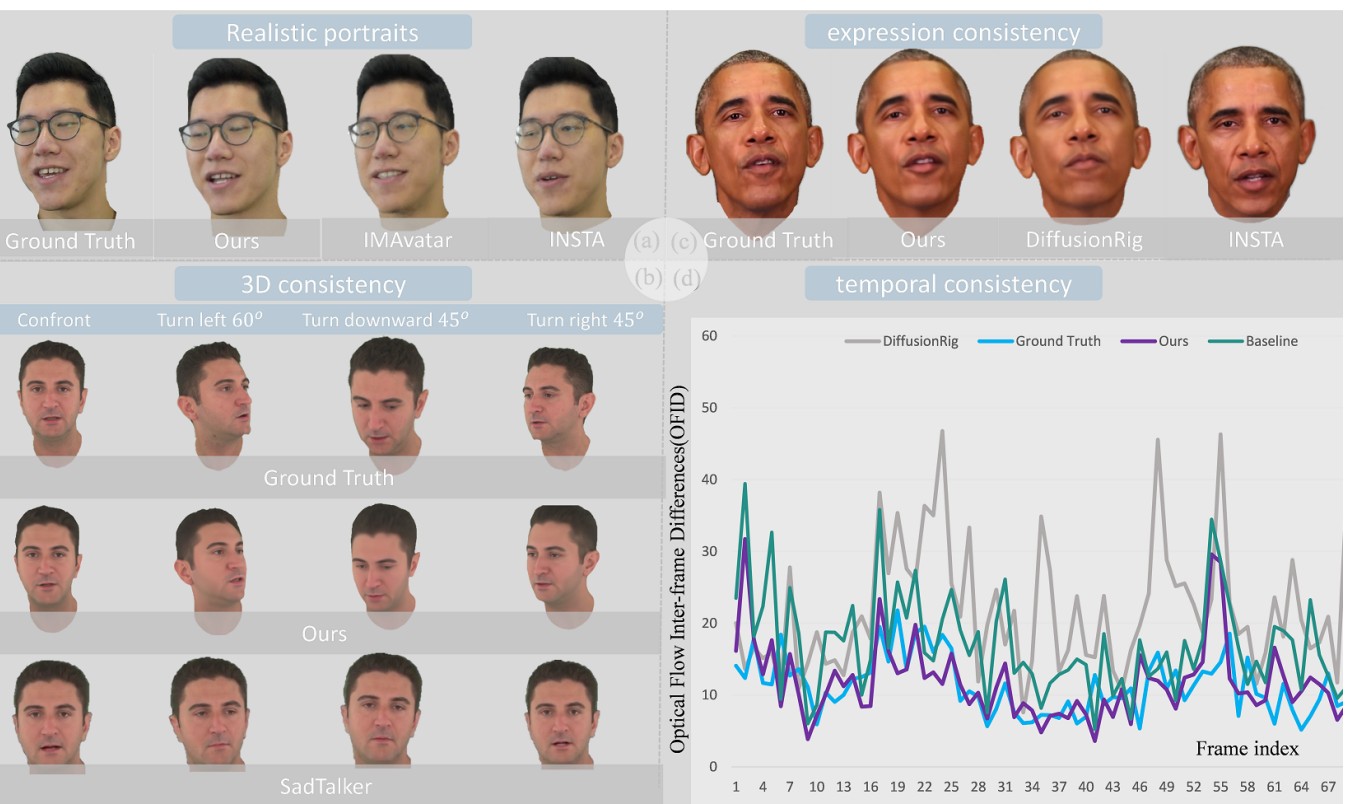

**Figure 1: For short monocular RGB videos, our method synthesizes avatars demonstrating full consistency in 3D (Figure (b)), temporal (Figure (d)), and emotional aspects (Figure (c)), while maintaining high quality (Figure (a)). Compared to state-of-the-art methods, ours effectively addresses all three consistency issues simultaneously. Notably, for temporal consistency, we quantify the degree of change between adjacent frames using optical flow. Comparative analysis with real data shows significant mitigation of temporal inconsistencies by our method. For the final video demonstration and details about optical flow, please refer to the supplementary materials.**

## ABSTRACT

Diffusion models have shown impressive potential on talking head generation. While plausible appearance and talking effect are achieved, these methods still suffers from temporal, 3D or expression inconsistency due to the error accumulation and inherent limitation of single-image generation ability. In this paper, we propose ConsistentAvatar, a novel framework for fully consistent and high-fidelity talking avatar generation. Instead of directly employing multi-modal conditions to the diffusion process, our method learns to first model the temporal representation for stability between adjacent frames. Specifically, we propose a Temporally-Sensitive Detail (TSD) map containing high-frequency feature and contour that vary significantly along time axis. Using a temporal consistent diffusion module, we learn to align TSD of the initial result to that of the video frame ground truth. The final avatar is generated by a fully consistent diffusion module, conditioned on the aligned TSD, rough head normal, and emotion prompt embedding. We find that the aligned TSD, which represents the temporal patterns, constrains the diffusion process to generate temporally stable talking head. Further, its reliable guidance complements the inaccuracy of other conditions, suppressing the accumulated error while improving the consistency on various aspects. Extensive experiments demonstrate that ConsistentAvatar outperforms the state-of-the-art methods on the generated appearance, 3D, expression and temporal consistency.

## CCS CONCEPTS

• **Computing methodologies** → *Reconstruction.*

## KEYWORDS

3D head avatars, Diffusion model, Consistent avatars

# 1 INTRODUCTION

In the realm of virtual reality and its associated applications, the creation of lifelike and animatable character avatars poses a significant challenge. Early approaches [5, 7, 34, 47, 49] often involve generating head avatars by fitting 3D Morphable Models (3DMMs) [2], utilizing a parameterized representation to describe the shape, texture, and details of the human face. By adjusting these parameters, precise control over facial features can be achieved, thereby demonstrating relatively strong 3D controllability. However, 3DMMs typically rely on certain assumptions, such as linear shape spaces, which may not fully capture the complex variations in facial shape, texture, and details. Consequently, the resulting facial renderings may lack realism (see Fig. 1 (a)).

To address these issues, some methods [11, 16, 36, 39, 45, 46] combine StyleGANs [15] with the priors of 3DMMs to improve the fidelity of generated avatars. For example, SadTalker [45] generates 3D motion coefficients of 3DMMs from audio and implicitly adjusts a novel 3D-aware face rendering. Thanks to the powerful generative capabilities of GANs, these methods often yield realistic images. However, these methods fundamentally rely on a single-image 2D generation process, where the image rendering process is intricately entangled and facial fitting capability is limited. Consequently, these methods encounter difficulties in achieving consistent 3D control (see Fig. 1 (b)). To overcome the limitations of 2D GAN-based methods, recent researches have tended to utilize Neural Radiance Field (NeRF) or similar implicit representations to generate face avatars [8, 17, 24, 26, 47], or integrate NeRF into GANs for 3D-aware dynamic face synthesis [17, 24, 26]. Typically, these methods learn dynamic NeRF based on input expressions as the conditions to represent the deformed 3D space. For example, IMAvatar [47] optimizes the implicit function on the mesh template to achieve more accurate 3D control. However, as the volume rendering involves manual approximation on the real-world imaging process, it has potential to lose fidelity on the rendered images. Additionally, NeRF struggles to decouple temporal information from 3D representation, resulting in temporal inconsistencies in such methods. As illustrated in Fig. 1 (a), INSTA [49] and IMAvatar [47] suffer from inaccurate expressions or appearance degradation. With the development of diffusion models, more recent studies [12, 18, 19, 35] have adopted this strategy to enhance the quality of face generation and editing. For example, DiffTalk [35] models the generation of talking heads as a denoising process driven by audio. Due to the powerful generation capabilities of diffusion models, these methods often achieve highly realistic results. However, these methods are essentially 2D-aware, so that they also lack sufficient 3D consistency.

Based on the above discussion, it seems that using more widely-covered conditions lead to fully consistent generated avatars in diffusion models. Actually, several efforts [6, 21] have integrated 3D-aware conditions like face normal or depth to guide their diffusion models. To further analyse relative strategies, we build a baseline diffusion model to generate avatars, conditioned on the low-resolution face image / normal generated from pre-trained INSTA [49] and emotion label embedding from CLIP [30]. As illustrated in Fig. 1 (d), we observe that DiffusionRig [6] and our baseline model both suffer from high temporal error and instability,

revealing that employing more conditions for diffusion models may not always lead to superior results. We argue the reason behind is two-fold: 1) the inaccuracy within these conditions disturbs the diffusion process and accumulates to the final result; 2) the based diffusion model [33] is image but not video generation model, and thus temporal consistency cannot be modeled or guaranteed. Note that, given a video as training data, the ground truth of temporal patterns has been implicitly contained. This motivate us to learn these patterns and constrain other noisy conditions for talking avatar generation.

To this end, we introduce ConsistentAvatar, a novel framework that combines diffusion models with the temporal, 3D-aware, and emotional conditions, for generating fully consistent and high-fidelity head avatars. Instead of directly generating avatar, we learn to first generate temporally consistent representation as a constraint to guide other conditions. Concretely, our method starts from the efficient INSTA method [49] and utilize its outputs as the initial results. In order to model the temporal consistency, we propose a Temporally-Sensitive Detail (TSD) generated by Fourier transformation, containing high-frequency information and contours that change significantly between frames. We extract TSD from coarse RGB output of INSTA and the target video frame, and propose a temporal consistency diffusion model to align the input TSD to the precise one. Besides, we utilize the coarse normal output of INSTA as the 3D-aware condition, and propose an emotion selection module to generate emotion embedding for each frame in an unsupervised manner. With the aligned TSD, normal, and emotion embedding as conditions, we then propose a fully consistent diffusion model to generate the final avatars. In this way, we guarantee the temporal consistency during the avatar generation, and complement other conditions to contribute to a fully consistent and high-quality avatar generation result. In summary, our contributions are as follows:

- We propose ConsistentAvatar, a diffusion-based neural renderer that generates temporal, 3D, and expression consistent talking head avatars.
- We learn to align a novel Temporally-Sensitive Details (TSD) to maintain the stability between generated frames, and complement rough normal and emotion conditions for high-fidelity generation.
- Extensive experiments demonstrate that ConsistentAvatar outperforms the state-of-the-art methods on the generated appearance quality, details, expression and temporal consistency.

# 2 RELATED WORK

## 2.1 3D Face Animation

Early methods [4, 32, 45] use 3DMM priors to instruct the generator. SadTalker [45], for instance, generates head poses and expressions of 3DMMs from audio and implicitly modulate a 3D-aware face synthesis model. Recent methods [1, 8, 10, 27, 47, 49], combined 3D representation techniques with 3DMMs to control the expressions and poses of avatars. For example, NHA [10] and IMAvatar [47] refine the mesh topology of FLAME [22] to achieve more realistic mesh-based avatars. INSTA [49] and RigNeRF [1] utilize 3DMMs to construct radiance fields. Additionally, other 3D representation

methods , such as GaussianAvatars [29], construct dynamic 3D representations based on 3D Gaussian splats rigged to a parametric morphable face model. More recent works [6, 19, 21, 35] have leveraged the powerful generation capabilities of 2D diffusion models. For instance, DiffusionAvatars [21] utilizes the 3D priors of the recent Neural Parametric Head Model (NPHM) [9] combines with a 2D rendering network for high-quality image synthesis. In our work, we integrate 3D representation with diffusion models, simultaneously leveraging 2D and 3D priors for consistent head avatar synthesis.

## 2.2 Controllable face generation with 2D Diffusion

Denoising Diffusion Probabilistic Models (DDPMs) [13] integrate image generation with a sequential denoising process of isotropic Gaussian noise. In this process, the model is trained to anticipate noise levels from the input image. Due to their remarkable ability to generate 2D content, they have gradually found application in face-related tasks. Many studies [6, 19, 21, 35], refine pre-trained diffusion models by introducing additional control factors like landmarks or depth. For example, DiffusionRig [6] suggests conditioning the diffusion model on rasterized grids, considering factors such as normals and albedo. Conversely, DiffTalk [35] utilizes audio-driven and landmark-based conditioning to improve identity generalization. Despite the good controllability and visual quality of these methods, they often lack accurate 3D representations or precise 2D detail priors. As a result, they struggle to maintain consistent 3D rendering across various viewpoints and temporal consistency in video rendering simultaneously.

## 2.3 Emotional Talking Video Portraits

Undoubtedly, emotions play a crucial role in shaping the authenticity of the synthesized portrait. Recently, there has been a growing effort to incorporate emotions into the synthesis process for better control over the final output. For instance, EMOCA [3] introduces a novel deep perceptual emotion consistency loss, ensuring alignment between the reconstructed 3D expression and the depicted emotion in the input image. Meanwhile, GMTalker [41] proposes a Gaussian mixture-based Expression Generator (GMEG), enabling the creation of a continuous and multimodal latent space for more versatile emotion manipulation. However, approaches like EVP [14] and EMMN [38], which directly infer emotions from labeled audio, may encounter accuracy issues due to the inherent complexity of emotional expression. In contrast, methods such as MEAD [40] and the work by Sinha et al. [37] implicitly learn the intrinsic relationship between emotions and facial expressions through the use of emotion labels. In our methodology, we leverage the MEAD [40] dataset to assign emotion labels to the experimental dataset based on emotional similarities, facilitating precise control over the emotions of the resulting portraits.

## 3 PRELIMINARY

**INSTA.** INSTA [49] is based on a dynamic neural radiance field composed of neural graphics primitives embedded around a parametric face model. It is capable of reconstructing photorealistic

digital avatars instantaneously in less than 10 minutes, while allowing for interactive rendering of novel poses and expressions. We choose INSTA to get a proxy of talking head under a target expression and head pose. The proxy, i.e., initial result used as input for our method is $512 \times 512$ rendered by volume rendering process, containing limited appearance quality and expression accuracy. Our method is capable of lifting the proxy to a high-fidelity and consistent avatar.

**Denoising Diffusion Probabilistic Models.** Denoising Diffusion Probabilistic Models (DDPMs) [13, 25, 31] belong to a class of generative models that take random noise images as input and progressively denoise them to produce photorealistic images. This process can be viewed as the reverse of the diffusion process, which gradually adds noise to images. The core component of DDPMs is a denoising network denoted as $f_\theta$, During training, it receives a noisy image $x_t$ and a timestep $t$ ($1 \leq t \leq T$), and predicts the noise at time $t : \epsilon_t$. More formally, the predicted noise at time $t$ is $\hat{\epsilon}_t = f_\theta (x_t, t)$, where $x_t = \alpha_t x_0 + \sqrt{1 - \alpha_t^2} \epsilon_t$, $\epsilon_t$ is a random, normally distributed noise image, and $\alpha_t$ is a hyperparameter that gradually increases the noise level of $x_t$ with each step of the forward process. The loss is computed based on the distance between $\epsilon_t$ and $\hat{\epsilon}_t$. Therefore, the trained model can generate images by taking a random noise image as input and progressively denoising it to achieve photorealism.

## 4 METHOD

In this section, we introduce ConsistentAvatar, a framework specifically designed for generating high-quality avatars while maintaining full consistency. An overview of this framework is illustrated in Fig. 2. Ensuring temporal consistency in generating portraits remains a challenge as the diffusion model struggles to directly learn time-related information from images. Therefore, we learn to generate temporally consistent representations as constraints to guide other conditions. We define Temporally-Sensitive Details (TSD), a detail map $I_{TSD}$ generated by Fourier transformation from the coarse RGB output of INSTA [49] and the target video frame. Furthermore, our experiments reveal that directly using the extracted TSD as a condition does not yield satisfactory temporal consistency, we propose a temporal consistency diffusion model to align the input TSD to the precise one (Sec. 4.1). After ensuring temporal consistency during the avatar generation process, we utilize the coarse normal output of INSTA and emotional text embeddings as conditions to construct a fully consistent diffusion model (Sec. 4.2). Finally, to expedite and further optimize our model, we draw inspiration from LCM [23] and SDXL [28], significantly reducing the necessary inference steps and further enhancing the quality of image generation (Sec. 4.3).

## 4.1 Temporally Consistent Module

As mentioned above, to ensure temporal consistency in generating portraits, we extract Temporally-Sensitive details (TSD) generated by Fourier transformation from the coarse RGB output of INSTA [49] and the target video frame. We then align these details through a diffusion model. This approach complements other conditions and contributes to achieving fully consistent and high-quality avatar generation results.

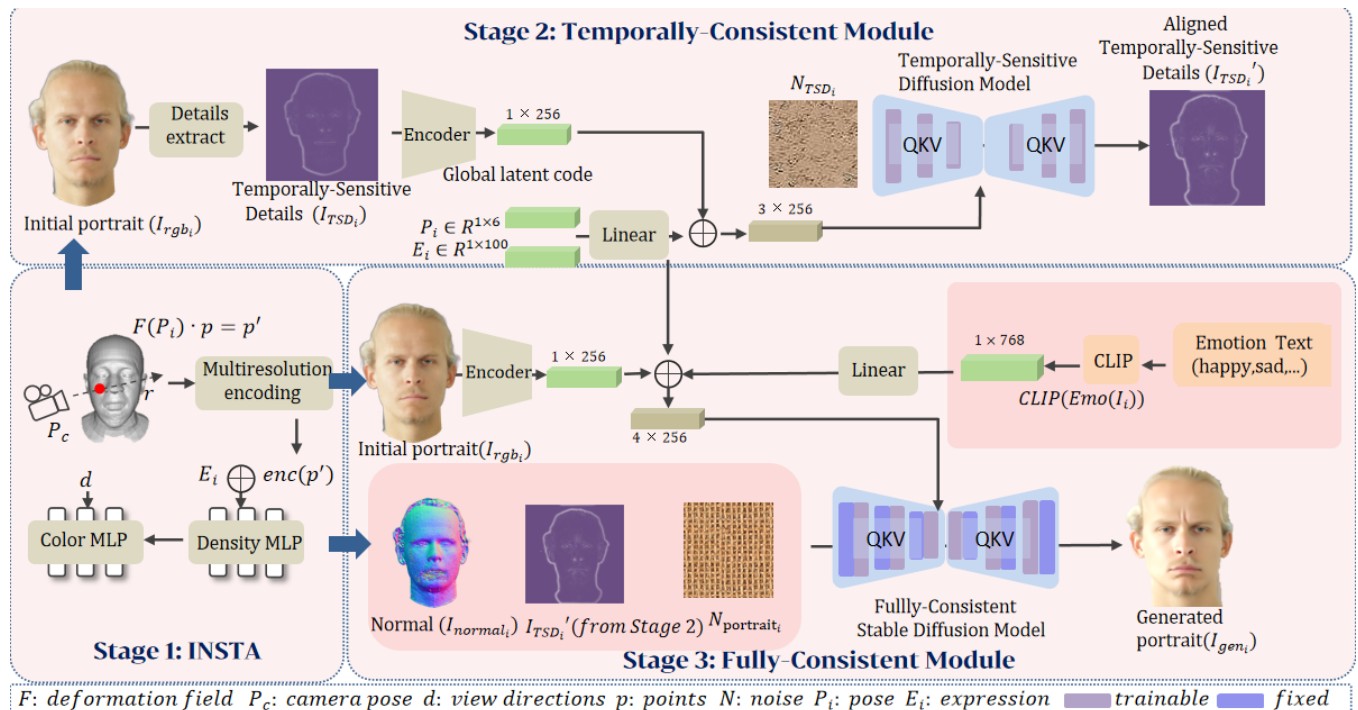

**Figure 2: Overview. ConsistentAvatar begins with the implementation of the highly efficient INSTA [49] method, leveraging its outputs as initial results (Stage 1). To address temporal consistency, we introduce a concept termed as Temporally-Sensitive Detail (TSD), derived through Fourier transformation. Extracting TSD from the coarse RGB output of INSTA and the target video frame, we develop a temporal consistency diffusion model to accurately align the input TSD with the precise one (Stage 2). Subsequently, we employ the coarse normal output of INSTA as a parameter for 3D perception and introduce an emotion selection module to generate emotion embeddings for each frame. By integrating aligned TSD, normal, and emotion embeddings as conditioning factors, we propose a fully consistent diffusion model to generate the final avatars (Stage 3).**

Given a monocular RGB video containing $K$ frames $\{I_1, I_2, \ldots, I_K\}$, camera pose $P_c$, tracked FLAME [22] meshes $M = \{M_i\}$ with corresponding expressions $E = \{E_i\} \in \mathbb{R}^{1 \times 100}$ and poses $P = \{P_i\} \in \mathbb{R}^{1 \times 6}$. We obtain RGB output and normal using INSTA [49]. The process is described as follows:

$$\mathcal{F}_{insta}(I_i, P_c, M_i, E_i, P_i) \rightarrow I_{rgb_i}, I_{normal_i}, \qquad (1)$$

where $\mathcal{F}_{insta}$ is the INSTA model. After obtaining the RGB output, we utilize Fourier transform and its inverse transform to extract TSD from the RGB output. We apply the Fourier transform to shift an image from its spatial domain to the frequency domain, revealing various frequency components. Retaining these high-frequency elements enables the Fourier transform to efficiently extract image details, which are represented as:

$$F_i = \int_{-\infty}^{+\infty} I_{rgb_i} e^{-jwt} dt = \begin{cases} = 0, & w \notin W \\ \neq 0, & w \in W \end{cases}, \qquad (2)$$

where the equation represents content extraction at frequency $w$ from the image $I_{rgb_i}$, with $e^{-jwt}$ as the orthogonal basis and $W$ as the frequency set of the image. Experimentally, this paper sets $w=10$. Then performing inverse Fourier transform to convert the

frequency domain back to images, which are represented as:

$$I_{TSD_i} = \int_{-\infty}^{+\infty} F_i e^{jwt} dt. \qquad (3)$$

As illustrated in Fig. 2, the TSD of a video frame contains high-frequency information and contours that represent crucial feature of expression, head pose and details for generating the avatar. However, TSD also varies significantly between adjacent frames. This inspires us to predict stable TSD of a frame from inaccurate $I_{TSD_i}$. Performing the same operation on the ground truth video frames allows us to obtain the ground truth of TSD. Therefore, we propose employing a Temporally-Sensitive Diffusion Model (TSDM) to align $I_{TSD_i}$. Specifically, inspired by [6], we first encode the obtained $I_{TSD_i} \in \mathbb{R}^{512 \times 512 \times 3}$ to obtain the global latent code $\mathcal{E}(I_{TSD_i}) \in \mathbb{R}^{1 \times 256}$. Then we pass $P_i$ and $E_i$ through a linear layer to obtain the same scale as $\mathcal{E}(I_{TSD_i})$ and concatenate them to get $f_c \in \mathbb{R}^{3 \times 256}$. We add new cross-attention layers to the U-Net, following IPAdapter [43]. Let $Z$ be the intermediate feature map computed by an existing cross-attention operation in the pre-trained LDM [33]: $Z = ATTENTION(Q, K, V)$. Then, we perform direct conditioning by adding another cross-attention layer:

$$Z \leftarrow Z = ATTENTION(Q, W^k f_c{}^t, W^v f_c{}^t). \qquad (4)$$

The more specific denoising process can be represented as follows:

$$\hat{\epsilon}_t = \mathcal{F}([x_t, f_c{}^t], t, x_0), \tag{5}$$

where $x_t$ is the noisy image at timestep $t \in \{1, 2, \ldots, T\}$, $x_0$ is the ground truth latent image, $\hat{\epsilon}_t$ is the predicted noise, and $\mathcal{F}$ represents the denoising model. The optimization objective of the entire denoising process is as follows:

$$\mathcal{L}_{TSD} = \|\hat{\epsilon}_t - \epsilon_t\|_2^2. \tag{6}$$

$\tilde{x}_{t-1} = x_t - \hat{\epsilon}_t$ is the denoising result of $x_t$ at time step t. The final denoised result $\tilde{x}_0$ is then upsampled to the pixel space with the pretrained image decoder $I'_{TSD_i} = \mathcal{D}_{TSD}(\tilde{x}_0)$, where $I'_{TSD_i} \in \mathbb{R}^{512 \times 512 \times 3}$ is the reconstructed TSD image. In this way, we align $I_{TSD_i}$ to an accurate one, providing the key information of real portrait changes along time axis. Further, $I'_{TSD_i}$ can be used as a reliable guidance for temporally-consistent avatar generation.

## 4.2 Fully Consistent Module

In Sec. 4.1, after passing TSD through the diffusion model, we obtain precise and temporally stable TSD. This compensates for other conditions that may not be particularly accurate. Therefore, by adding normal condition and emotion condition, we alleviate 3D consistency and expression consistency.

**Emotion Condition:** For emotion condition, we propose utilizing the MEAD [40] dataset to acquire emotion labels for our experimental dataset. As described in Fig. 3, the MEAD comprises numerous monocular video clips $\{V_1, V_2, \ldots, V_M\}$, each clips corresponds to an emotional label $L_m \in \{L_1, L_2, \ldots, L_M\}$ (such as anger, happy, etc.). Leveraging this rich emotion dataset, we construct a facial expression database. Subsequently, employing DECA [7], we compute facial expression vectors for each frame in every clip $E_{mn}$, $n$ represents the frame index of the clip. And then determine the similarity between the current frame's facial expression vector $E_i$ and those in the database. Finally, we assign the emotion label corresponding to the facial expression vector with the highest similarity in the database as the emotion label for the current frame. The process is described as follows:

$$\mathcal{EMO}(I_i) = L(argmax(cs(deca(I_i), deca(V_{mn})))), \tag{7}$$

where $I_i$ is an image from our dataset, and $V_{mn}$ is a frame from the MEAD dataset, $cs()$ represents the cosine similarity function, which is used to measure the similarity between two expression vectors. $argmax()$ retrieves the image corresponding to the maximum similarity, which belongs to the MEAD dataset. After acquiring emotional labels corresponding to image through $\mathcal{EMO}(I_i)$, we use the text encoder in CLIP [30] to extract textual features of emotion labels, obtaining the final emotion code $T_i \in \mathbb{R}^{1 \times 768}$. After acquiring STD, normal, and emotion conditions, we perform the same linear operation on $T_i$ as on $E_i$ and $P_i$, and then concatenate them to obtain $f_d \in \mathbb{R}^{4 \times 256}$. At that time, it can be utilized as a condition through cross-attention operation. Finally, we construct a Fully Consistent Stable Diffusion model (FCSD). Specifically, we utilize the pretrained Stable Diffusion model (SD) [33] as the backbone network to expedite training and enhance generation quality. Additionally, we incorporate a fully consistency module, effectively integrating consistency-related information into the backbone network, to construct a ControlNet [44]. The final rendering is achieved

through an iteratively denoising the full noise $x_T$ with our fully-consistent neural renderer $\mathcal{S}$ conditioned on $z_c$:

$$\hat{\epsilon}_t = \mathcal{S}([x_t, f_d^t], t, x_0, C(x_t, z_c)), \tag{8}$$

where C is the ControlNet architecture and $z_c$ is the concatenation of the normal condition and the TSD condition. During training, we minimize the following loss:

$$\mathcal{L}_{portrait} = \|\hat{\epsilon}_t - \epsilon_t\|_2^2. \tag{9}$$

Similar to Sec. 4.1, the final denoised result $\tilde{x}_0$ is then upsampled to the pixel space with the pretrained image decoder $I_{gen_i} = \mathcal{D}_{portrait}(\tilde{x}_0)$, where $I_{gen_i} \in \mathbb{R}^{512 \times 512 \times 3}$ is the reconstructed face image. This process provides the network with additional multimodal conditions, helping it control the 3D and expression consistency of the characters under the well-guided aligned TSD, thereby assisting in generating portrait animations with more precise motion.

## 4.3 Optimization

It is indisputable that both generation speed and the quality of portrait generation are pivotal criteria for task assessment. We closely follow the development of diffusion models and, based on this foundation, further accelerate efficiency and enhance image quality.

**Latent Consistency Model (LCM).** To ensure our model maintains high-quality generation while minimizing the required steps, we employ LCM [23] to expedite the generation process. The core concept of this model lies in redefining the fundamental logic of traditional diffusion models (such as DDPM [13]) in the generation process. Traditional diffusion models generate final results by gradually reducing noise in images, typically through iterative and time-consuming processes. In contrast, LCM [23] transforms traditional numerical ordinary differential equation (ODE) solvers into neural network-based solvers, enabling direct prediction of the final clear image and thereby reducing intermediate steps, significantly enhancing efficiency. By employing LCM [23] to enhance our model, we have reduced the required steps for predicting the final image from around 1000 steps to approximately 10 steps.

**Refiner.** Inspired by Stable Diffusion XL (SDXL) [28], we have further elevated the generation quality of our model. SDXL represents the latest optimized version of Stable Diffusion, comprising a two-stage cascaded diffusion model consisting of a Base model and a Refiner model. Integrating the Refiner model with our own, after the Base model generates latent features of the image, we utilize the Refiner model to conduct minor noise reduction and enhance detail quality on these latent features.

# 5 EXPERIMENT

## 5.1 Setup

**Dataset:** In our experiments, we utilize four datasets. Initially, we employ the dataset released by INSTA [49] to train the primary framework. This dataset comprises 10 monocular videos, each capturing the performance of an individual actor. These videos undergo cropping and resizing, achieving a resolution of $512^2$, effectively removing extraneous elements from the facial region through background subtraction. Additionally, for equitable comparison with

Figure 3: Emotion text selection module diagram.

other methods, we utilize two datasets from IMAvatar [47] and Ner-Face [8] which share the same data format as INSTA [49]. Finally, during the emotional labeling phase, we utilize the Multi-view Emotional Audio-visual Dataset (MEAD) [40], which includes a dialogue video corpus featuring 60 actors conversing with 8 different emotions. High-quality audio-visual clips from seven different perspectives are captured in a strictly controlled environment.

**Evaluation Protocol:** We utilize the final 350 frames of each video in our dataset for testing purposes, while the remaining frames are allocated for training. To assess image quality, we employ metrics consistent with state-of-the-art methods [47, 49], including L2 loss, structural similarity index (SSIM), peak signal-to-noise ratio (PSNR), and the perceptual metric LPIPS. Additionally, to evaluate the temporal consistency of the generated videos, we utilize optical flow to compute the degree of change between adjacent frames.

**Implementation Details:** Our framework was implemented using PyTorch on a machine equipped with an RTX 3090 GPU. The training process is divided into three stages. In the first stage, all experimental configurations match those of INSTA [49], yielding RGB and normal outputs for the portraits. Moving to the second stage, we utilize the temporally unstable TSDs obtained from the RGB outputs of the first stage to train an TSDs generator with ground truth supervision. We utilize the Adam optimizer [20] with a learning rate of $10^{-5}$ and conduct 3000 iterations with a batch size of 4. Finally, in the third stage, we train a fully consistent portrait generator based on the outputs of the first two stages. Once again, we use the Adam optimizer with a learning rate of $10^{-4}$ and conduct 15000 iterations with a batch size of 4.

**Baseline setting:** To illustrate the role of TSD in temporal consistency more clearly, we use Stage 1 combined with Stage 3 as our baseline. Here, TSD is not learned through Stage 2, but is directly used as a condition for ControlNet [44].

## 5.2 Comparison with the State-of-the-art Methods

**Image quality evaluation:** We conduct comprehensive experiments on our dataset, assessing the quality and consistency of the synthetic digital human avatars generated by our method, and comparing them with state-of-the-art methods such as IMAvatar [47], DiffusionRig [6] and INSTA [49]. As shown in Fig. 4, our approach yields more realistic results. We capture facial details such as wrinkles and eyeglass frames well. In contrast, other methods

| Method | Dataset | L2 ↓ | PSNR ↑ | SSIM ↑ | LPIPS ↓ | Time(s) ↓ |
|---|---|---|---|---|---|---|
| NHA [10] | INSTA | 0.0022 | 27.71 | 0.95 | 0.040 | 0.63 |
| NeRFace [8] | | 0.0018 | 29.28 | 0.95 | 0.070 | 9.68 |
| IMAvatar [47] | | 0.0023 | 27.62 | 0.94 | 0.060 | 12.34 |
| MAVavatar [42] | | 0.0027 | 25.76 | 0.93 | 0.070 | 1.10 |
| INSTA [49] | | 0.0018 | 28.97 | 0.95 | 0.050 | **0.05** |
| DiffusionRig [6] | | 0.0016 | 31.34 | 0.96 | 0.047 | 4.10 |
| **w/o stage2(Baseline)** | | 0.0010 | 33.78 | 0.97 | 0.040 | 2.23 |
| **w/o LCM** | | 0.0008 | 34.05 | 0.97 | 0.038 | 8.20 |
| **Ours** | | **0.0008** | **34.05** | **0.97** | **0.038** | 2.23 |
| PointAvatar [48] | PointAvatar | 0.0027 | 26.04 | 0.88 | 0.147 | 0.80 |
| **w/o stage2(Baseline)** | | 0.0012 | 32.42 | 0.93 | 0.044 | 2.23 |
| **w/o LCM** | | 0.0011 | 32.72 | 0.95 | 0.040 | 8.20 |
| **Ours** | | **0.0011** | **32.72** | **0.95** | **0.040** | 2.23 |
| NeRFace [8] | NeRFace | 0.0016 | 26.85 | 0.95 | 0.060 | 9.68 |
| **w/o stage2(Baseline)** | | 0.0014 | 32.80 | 0.96 | 0.045 | 2.23 |
| **w/o LCM** | | 0.0010 | 33.35 | 0.96 | 0.040 | 8.20 |
| **Ours** | | **0.0010** | **33.35** | **0.96** | **0.040** | 2.23 |

**Table 1: Quantitative analyses between our method and state-of-the-art models.**

like INSTA [49] fail to capture wrinkle information and other details. Tab. 1 presents quantitative results, demonstrating that our method achieves state-of-the-art performance across three different datasets. Particularly noteworthy is the substantial improvement in both PSNR and L2 loss compared to INSTA [49]. Taken together, these metrics indicate that our generated portraits are closer to ground truth and more realistic. Additionally, our ability to better capture facial expressions contributes to the overall enhancement of the results. Additional. We are well aware of the importance of real-time performance, so as mentioned in Sec. 1, we have focused on improving speed by introducing LCM [23] on top of our model. As shown in the data in the Tab. 1, we have significantly reduced the time. Note that the above results were obtained with a time step setting of 10 during the denoising process, while the default diffusion model step size is set to 1000. The incorporation of LCM [23] significantly reduces the time step.

**3D consistency results:** 3D consistency measures whether the character maintains the correct appearance from different viewpoints. As shown in Fig. 1 (b), methods like SadTalker [45] do not naturally exhibit good 3D consistency since they do not involve 3D operations during learning and rely on a single-image 2D generation process. From Fig. 5, we can also observe the advantage of our method, with the angles of head rotation being closer to the ground truth and the results appearing more realistic. For quantitative results, we utilize DECA [7] to estimate the pose coefficients of the generated portraits, calculating the error between the predicted pose coefficients and the ground truth, referred to as Pose Error

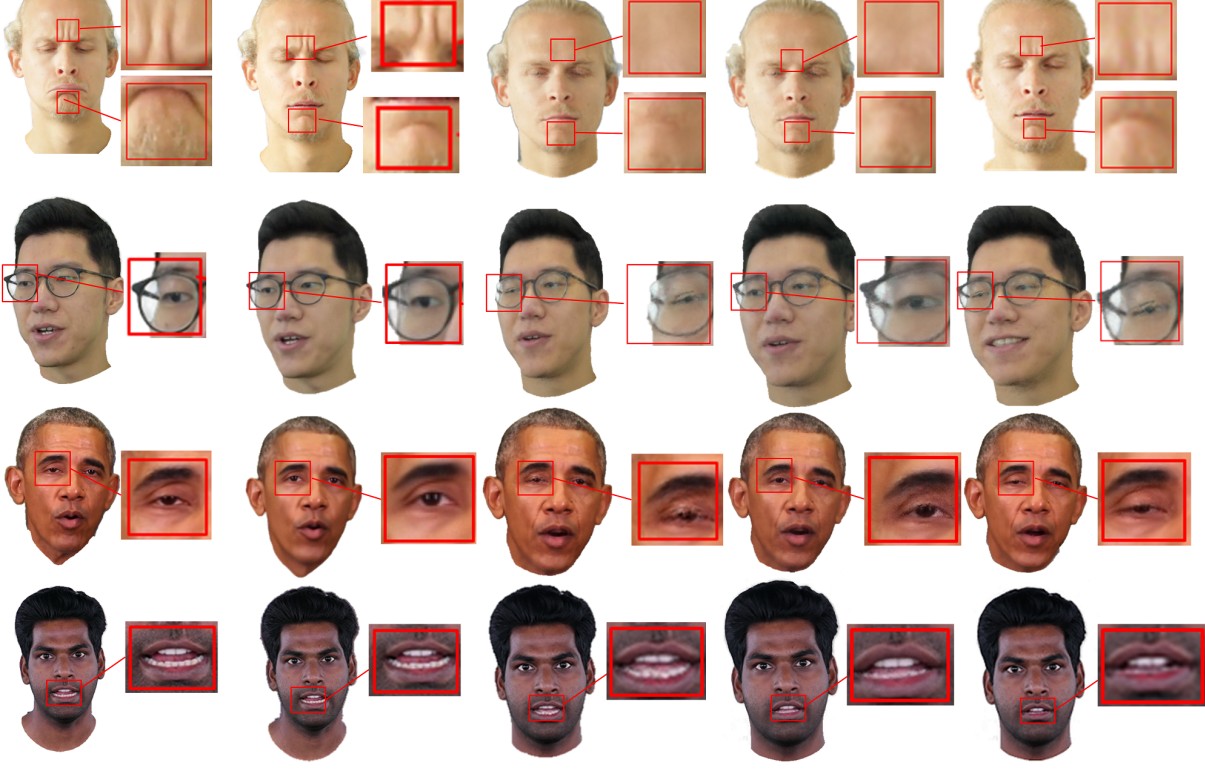

Ground truth          Ours          INSTA          PointAvatar          IMAvatar

**Figure 4: Qualitative Results. Clearly, the facial avatars reconstructed by our method exhibit accurate and lifelike details, including intricate features such as wrinkles and eyes. Other methods produce excessively smooth results.**

| Method | PE ↓ | Method | EE ↓ |
|---|---|---|---|
| SadTalker [45] | 0.0755 | INSTA [49] | 1.9236 |
| w/o aligned TSD | 0.02328 | Diffusionrig [6] | 1.7665 |
| w/o normal | 0.03653 | w/o aligned TSD | 1.4355 |
| **Ours** | **0.0096** | w/o emotion text | 1.1922 |
| – | – | **Ours** | **0.8452** |

**Table 2: Quantitative results and ablation results compared to state-of-the-art methods.**

(PE), measured by the Euclidean distance. As shown in Tab. 2, our method significantly outperforms SadTalker, with minimal error compared to the ground truth.

**Expression consistency results:** The expression consistency measures how well the generated results fit the target expressions. As shown in Fig. 6, our method is capable of achieving more accurate expressions. For quantitative analysis, similar to PE, we employ DECA [7] to evaluate the expression coefficients of the generated portraits, calculating the disparity between the predicted expression coefficients and the ground truth, referred to as Expression Error (EE). As depicted in Tab. 2, our approach surpasses both Diffusionrig [6] and INSTA [49], exhibiting minimal error relative to the ground truth.

**Temporal consistency results:** The evaluation of temporal consistency evaluates the stability of the generated video, focusing on the absence of jitter or significant fluctuations between frames, ensuring smooth playback. We compare our method with state-of-the-art approaches like INSTA[49], SadTalker[45] and DiffusionRig [6] in various categories. We utilize optical flow to compute the degree of change between two frames. Naturally, we consider the normal variation between video frames. Therefore, we compare it with the ground truth as well. From the Fig. 7, it's evident that our approach closely approximates the ground truth.

## 5.3    Ablation Studies

We conduct a series of ablation studies to analyze the different components of our method. Specifically, we focus on 1) the impact of unaligned coarse TSD versus aligned TSD through the diffusion model on other conditions and on temporal consistency; 2) the impact of the emotion condition on the final results; 3) the impact of the normal condition on the final 3D consistency;

**Aligned TSD and Unaligned TSD.** From Fig. 1 (d), it is evident that the temporal consistency of our baseline is inferior to that of our final method. Since the baseline directly adopts unaligned TSD conditions, this proves that aligned TSD can provide better

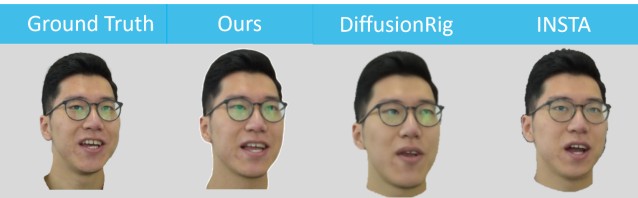

**Figure 5: Comparison with SadTalker in terms of 3D consistency.**

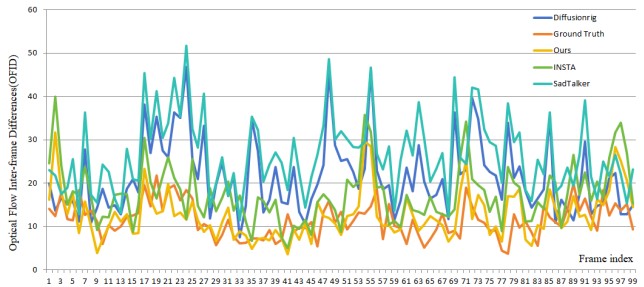

**Figure 6: Comparison with different state-of-the-art methods in terms of expression consistency.**

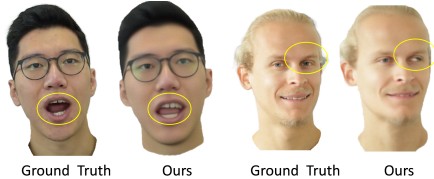

**Figure 7: Comparison with different state-of-the-art methods in terms of temporal consistency.**

temporal consistency. From Fig. 8 (a) and (c), it can be observed that aligned TSD conditions offer more accurate expression and better 3D consistency.

**Emotion Condition.** With the aligned TSD as a condition, we compared the results with and without the addition of the emotion condition. As shown in Fig. 8 (b), when the emotion condition is included, it helps the model fine-tune facial expressions, resulting in outcomes closer to the ground truth.

**Normal Condition.** With the aligned TSD as a condition, we compare the results with and without the addition of the normal condition. As shown in Fig. 8 (d), when the normal condition is not included, it fails to maintain good 3D consistency, resulting in artifacts and other noise.

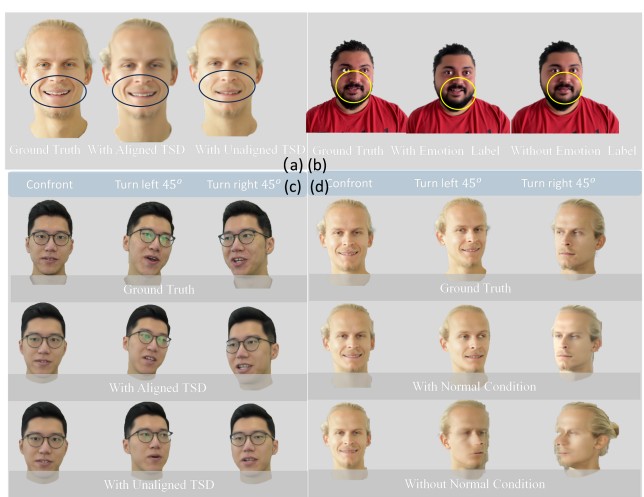

**Figure 8: The results of the ablation experiments on TSD condition, normal condition, and emotion condition.**

## 6 LIMITATION:

Although our work achieves realistic and fully consistent portraits through the diffusion model and the utilization of TSD, normal, and emotion conditions, there are still limitations. Our method often lacks accurate modeling of teeth and eyeball due to the lack of geometric constraints, as depicted in Fig. 9.

**Figure 9: The results of our method's limitations.**

## 7 CONCLUSION

ConsistentAvatar presents a novel framework for generating talking avatars with full consistency and high fidelity. We introduce a Temporally-Sensitive Detail (TSD) map containing high-frequency features and contours that exhibit significant variation over time. Utilizing a temporal consistent diffusion module, we align the TSD of the initial result with the ground truth of the video frame. The final avatar is generated using a fully consistent diffusion module, conditioned on the aligned TSD, rough head normal, and emotion prompt embedding. Aligned TSD, representing temporal patterns, guides the diffusion process to produce temporally stable talking heads. Its reliable guidance supplements the inaccuracies of other conditions, thereby reducing accumulated errors and enhancing consistency across various aspects.

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
