# OpenReview forum: "ConsistentAvatar: Learning to Diffuse Fully Consistent Talking Head Avatar with Temporal Guidance"
_acmmm.org/ACMMM/2024/Conference — MM2024 Oral_

### Official Review · Reviewer_uT5Z · 2024-05-22

**Rating:** 5
**Confidence:** 3

**Summary:**

The paper "ConsistentAvatar: Learning to Diffuse Fully Consistent Talking Head Avatar with Temporal Guidance" introduces a novel framework that tackles temporal, 3D, and expression inconsistencies in talking head avatar generation. By leveraging Temporally-Sensitive Details (TSD) and optical flow for temporal guidance, the method achieves significant improvements in consistency across various aspects. Comparative analysis with real data demonstrates the effectiveness of the proposed approach. The experimental results showcase superior performance compared to state-of-the-art methods in terms of appearance, 3D consistency, expression, and temporal consistency. The paper also provides clear implementation details and optimization techniques, such as the use of LCM for faster inference speed and the Refiner for improved detail quality. Overall, the paper makes a solid contribution to the field of avatar generation by addressing multiple consistency issues simultaneously and demonstrating promising results through extensive experiments.

**Strengths:**

1.	The paper introduces a novel framework, ConsistentAvatar, which addresses temporal, 3D, and expression inconsistencies in talking head avatar generation simultaneously.
2.	The method proposes a novel Temporally-Sensitive Details (TSD) module to maintain the stability between generated frames, which can further aid the rough normal and emotion conditions for high-fidelity avatar generation.
3.	The paper provides a comprehensive comparative analysis with real data, demonstrating significant mitigation of temporal inconsistencies by the proposed method. Extensive experiments showcase that the proposed method outperforms state-of-the-art methods in terms of appearance, 3D consistency, expression, and temporal consistency.
4.	The paper provides clear implementation details, including the training process stages and optimization techniques like LCM and Refiner which improve efficiency and image quality respectively.

**Limitations:**

Weaknesses
1.	The paper demonstrated about the improvement on generation speed by the incorporation of LCM in Table 1. However, apart from the generation quality, rendering speed is also crucial as stated in the paper. The proposed method seems cannot reach the real-time requirement. More discussion needs to be addressed to reach the goal.
2.	The paper may benefit from discussing potential real-world applications or scenarios where the ConsistentAvatar framework could be utilized.

Questions
1.	How does the ConsistentAvatar framework handle the trade-off between computational efficiency and maintaining high-quality avatar generation results, especially when incorporating the LCM and Refiner?
2.	The LCM can generate high-quality results with faster speed, however, sometimes the time step might be not enough for generating a realistic result, how does ConsistentAvatar address this issue?

Limitations
The paper acknowledges limitations in accurately modeling teeth and eyeballs due to the lack of geometric constraints. These limitations might impact the overall realism and fidelity of the generated avatars. Are there potential avenues for improvement in future research to address these challenges?

**Suitability:**

3

---

### Official Review · Reviewer_K1X4 · 2024-05-24

**Rating:** 3
**Confidence:** 3

**Summary:**

This paper focuses on talking head generation and introduces a diffusion-based neural render method (namely ConsistentAvatar) with high temporal, 3D and expression consistency. Specifically, this paper suggests a Temporally-Sensitive Detail (TSD) map containing high-frequency information that vary significantly in different frames. The authors inject TSD into diffusion models to improve the temporal stability. Comprehensive experiments showcase the superiority of the proposed method compared to state-of-the-art methods.

**Strengths:**

+ This paper in well organized and easy to understand.
+ The overall paper is technically sound.
+ The proposed method shows non-trivial performance improvement compared to several prior methods. The videos in supplementary material clearly show temporal consistency of the proposed method.

**Limitations:**

- The novelty of this paper is marginal. Most techniques are from existing methods, such INSTA [49], ControlNet [44] and LCM [23]. Only the usage of high-frequency information (TSD) to enhance temporal consistency is somewhat new in the talking head generation task. However, I cannot find the results of the proposed method without TSD in Table 1.
- Some design choices are confusing. I am wondering why stage 2 can improve the performance. Although the authors claim necessity of temporally consistent module (stage 2), this module should marginal performance improvement as shown in Table 1.
- In Table 1, I am wondering why the baseline (i.e., the final model without stage 2) has the same time costs.
- Some typos should be fixed, such as ‘We learn to align a novel Temporally-Sensitive Details’ in Line 211.

**Suitability:**

3

---

### Official Review · Reviewer_pNqa · 2024-05-25

**Rating:** 4
**Confidence:** 3

**Summary:**

This paper proposes ConsistentAvatar, a diffusion-based head avatar generation framework, which renders avatar videos initialized by a Nerf-based approach INSTA to generate high-fidelity talking face videos.
To overcome the inconsistency of diffusion models, this paper introduces Temporal-Sensitive Details (TSD), which extract high-frequency temporal information from head video by Fourier transformation and utilizes TSD as a condition for video generation.
To generate consistent avatar videos, ConsistentAvatar proposes a fully-consistent module, where multiple conditions, including normal map, TSD, and initial avatar videos are inputted into the diffusion model. Additionally, emotion text embedding is used to control the avatar's emotions.
Experiments demonstrate that ConsistentAvatar performs well on the generated appearance, 3D, expression, and temporal consistency.

**Strengths:**

The idea of generating temporally consistent representations as conditions is interesting. To achieve this idea, this paper extracts temporal high-frequency features as TSD, and the refined stable TSD is utilized as a condition for the diffusion model. The experiments demonstrate the effectiveness of TSD.
The full pipeline for avatar video generation is complete, where multiple conditions are considered for consistency, and emotion labels are embedded for control. Additionally, LCM and refiner are proposed for further accelerate efficiency and enhance image quality.
The provided videos in supplementary materials show better consistency than the diffusion-based method, diffusionrig.

**Limitations:**

The current experimental results should all be based on reconstruction. Could you compare the non-reconstructed poses as well?

In Table 2, why does the PE metric only compare with SadTalker, but not Insta, DiffusionRig, PointAvatar, or ImAvatar?

Although optical flow can be used to measure temporal consistency, can other common metrics like FVD also be used for evaluation?

**Suitability:**

2

---

### Official Review · Reviewer_uvEK · 2024-05-25

**Rating:** 2
**Confidence:** 4

**Summary:**

The paper addresses the challenges faced by existing diffusion models in generating talking heads, particularly issues related to temporal, 3D, and expression inconsistencies. These issues arise from error accumulation and the limitations inherent in single-image generation capabilities. To overcome these challenges, the authors propose ConsistentAvatar, a novel framework that ensures fully consistent and high-fidelity talking avatar generation.

**Strengths:**

1. The proposed Temporally-Sensitive Detail (TSD) map helps maintain stability between adjacent frames, reducing flickering and inconsistencies over time.
2. The use of rough head normal from INSTA and emotion prompt embeddings from CLIP, in conjunction with the TSD, results in more lifelike and expressive avatars.
3. ConsistentAvatar achieves a higher level of detail and expression accuracy in the generated avatars.

**Limitations:**

1. The baseline is a combination of INSTA and ControlNet, which is incremental. As shown in Table 1, the enhancement provided by TSD is not significant, limiting the overall contribution of this paper.
2.  ConsistentAvatar trains INSTA, a diffusion model, and a ControlNet. the efficiency of training is a problem. It is better to make the training efficiency clear.
3. The comparison in the ablation study is not convincing. More metrics should be presented to effectively evaluate the performance of different components, such as metrics reported in Table 1.
4. Poor writing. There are quite a few grammar mistakes and awkward phrases with unclear definitions. These problems make this paper harder to read and understand. For instance:
-Abstract (line 45): “these methods still suffer” -> “these methods still suffer”
-Abstract (line 54): “high-frequency feature and contour” -> “high-frequency features and contours”
-Abstract (line 55): “vary significantly along time axis” -> “vary significantly along the time axis”
-Introduction (line 141): “recent researches” -> “recent research” or “recent studies”
-Introduction (line 176): “the reason behind is two-fold”->”the reasons behind are two-fold”
-Introduction (line 182 and 183) “This motivate … constrain…”-> “This motivates … constrains…”
-Abstract (line 104): “the inaccuracy of other conditions” which kind of inaccuracy conditions?
-And so on…
5. The figures are a little messy, especially Figures 2 and 3. In Figure 3, the phrase “if x is…” is used, but “x” is not defined. The figures should be cleaned up to enhance the paper's readability and make it suitable for publication.
6. SadTalker is not a perfect baseline for this problem. More emotional talking-head generation methods[1,2] should be discussed.
[1] Emote Portrait Alive: Generating Expressive Portrait Videos with Audio2Video Diffusion Model under Weak Conditions. Ji et al. ACM SIGGRAPH 2022
[2] Efficient emotional adaptation for audio-driven talking-head generation. Gan et al. ICCV 2023

**Suitability:**

3

---

### Meta-Review · Area_Chair_A3gM · 2024-07-04

**Recommendation:** Accept (Oral)
**Confidence:** 4

**Metareview:**

The paper proposes a new method for high-fidelity talking head generation. Generally, I find this paper quite impressive to some extent $-$ (1) it provided sufficient qualitative evidence (both on paper and through the supplemental demo clips) which was quite convincing; (2) there is a measured amount of analysis and discussion afforded to the aspect of temporal consistency, which bodes well with the theme of the paper. It must be said that achieving a temporally consistent generation with diffusion models is a rather challenging task, and this paper sets out to address this critical issue. Comparisons with state-of-the-art methods were convincing and showed the promise of the proposed idea. It would be great if the authors could demonstrate the generation w.r.t emotions ($L_m$) and if the expression consistencies ($deca$) are better for certain emotion types than others.

Indeed, I concur with the reviewers' assessment that the authors did a thorough job of addressing their concerns in the rebuttal. I find the rebuttal concise yet meticulous. I hope the authors will incorporate these changes into the final version of the paper to increase the quality of the final paper.

I'm happy to recommend the acceptance of this paper for oral presentation.